# Possible Dietary Effects of Insect-Based Diets across Zebrafish (*Danio rerio*) Generations: A Multidisciplinary Study on the Larval Phase

**DOI:** 10.3390/ani11030751

**Published:** 2021-03-09

**Authors:** Matteo Zarantoniello, Basilio Randazzo, Gloriana Cardinaletti, Cristina Truzzi, Giulia Chemello, Paola Riolo, Ike Olivotto

**Affiliations:** 1Dipartimento di Scienze della Vita e dell’Ambiente, Università Politecnica delle Marche, via Brecce Bianche, 60131 Ancona, Italy; matteo.zarantoniello@gmail.com (M.Z.); b.randazzo.live@gmail.com (B.R.); c.truzzi@staff.univpm.it (C.T.); g.chemello@staff.univpm.it (G.C.); 2Dipartimento di Scienze Agro-Alimentari, Ambientali e Animali (Di4A), Università di Udine, via Sondrio 2/A, 33100 Udine, Italy; gloriana.cardinaletti@uniud.it; 3Dipartimento di Scienze Agrarie, Alimentari ed Ambientali, Università Politecnica delle Marche, via Brecce Bianche, 60131 Ancona, Italy; p.riolo@staff.univpm.it

**Keywords:** Black Soldier Fly, nutritional programming, zebrafish, insect meal, *Hermetia illucens*

## Abstract

**Simple Summary:**

Fish meal and fish oil represent the optimal ingredients for aquafeed formulation. However, their partial or complete substitution with more sustainable alternatives, like insects, is required for a further development of the aquaculture sector. Nutritional programming through parental feeding may enhance the ability of the progeny to utilize insect-based diets. In the present study, five experimental diets characterized by increasing fish meal substitution levels with full-fat Black Soldier Fly (*Hermetia illucens*; BSF) prepupae meal (0%, 25%, 50%, 75% and 100%), used for zebrafish broodstock rearing, were provided to the progeny (first filial generation, F1). The effects of BSF-based diets on F1 zebrafish larvae were investigated through a multidisciplinary approach. No significant differences among the experimental groups were observed in terms of growth, hepatic lipid accumulation and gut health. Furthermore, increasing fish meal substitution levels with BSF prepupae meal resulted in a positive modulation of both stress and immune response. Results demonstrated that nutritional programming via broodstock nutrition should be considered a valuable solution to increase the use of insect meal in aquafeeds formulation and improve fish culture sustainability.

**Abstract:**

Insects represent a valuable and sustainable alternative ingredient for aquafeed formulation. However, insect-based diets have often highlighted controversial results in different fish species, especially when high inclusion levels were used. Several studies have demonstrated that nutritional programming through parental feeding may allow the production of fish better adapted to use sub-optimal aquafeed ingredients. To date, this approach has never been explored on insect-based diets. In the present study, five experimental diets characterized by increasing fish meal substitution levels with full-fat Black Soldier Fly (*Hermetia illucens*; BSF) prepupae meal (0%, 25%, 50%, 75% and 100%) were used to investigate the effects of programming via broodstock nutrition on F1 zebrafish larvae development. The responses of offspring were assayed through biometric, gas chromatographic, histological, and molecular analyses. The results evidenced that the same BSF-based diets provided to adults were able to affect F1 zebrafish larvae fatty acid composition without impairing growth performances, hepatic lipid accumulation and gut health. Groups challenged with higher BSF inclusion with respect to fish meal (50%, 75% and 100%) showed a significant downregulation of stress response markers and a positive modulation of inflammatory cytokines gene expression. The present study evidences that nutritional programming through parental feeding may make it possible to extend the fish meal substitution level with BSF prepupae meal in the diet up to almost 100% without incurring the well-known negative side effects of BSF-based diets.

## 1. Introduction

The partial or complete replacement of fish meal (FM) and fish oil (FO) in aquafeed formulations represents an ongoing challenge in the aquaculture sector [1,2]. For this reason, improvements in the use of low-cost and more sustainable alternative ingredients are considered a priority for the further development of fish culture [3]. In light of the circular economy concept and the importance of by-product reuse, insects have gained great attention as a promising aquafeed ingredient due to their nutritional value, bio-converting efficiency and their low environmental requirements [4,5].

In particular, the Black Soldier Fly (*Hermetia illucens*; BSF) is able, during its larval development, to grow on organic by-products, converting them into valuable biomass with a nutritional composition dependent on the quantity and the quality of feed offered [6,7,8]. Due to its promising protein content and essential amino acid pattern, the use of different BSF dietary inclusion levels has been widely investigated in several farmed fish species, including rainbow trout (*Oncorhynchus mykiss*) [9,10], Atlantic Salmon (*Salmo salar*) [11,12] and Siberian sturgeon (*Acipenser baerii*) [13,14], as well as in experimental models like zebrafish (*Danio rerio*) [8,15,16]. In addition, as recently reviewed by Zarantoniello and collaborators [17], different diets in terms of BSF dietary inclusions, developmental stage (larvae or prepupae) and lipid content (full-fat, partially or totally defatted) have been tested in recent years on several fish species. However, the BSF fatty acid profile (characterized by high content of medium-chain saturated—SFA—and monounsaturated—MUFA—fatty acids, rather than long-chain polyunsaturated—PUFA—ones) [18] on fish growth, welfare and quality still deserve a deeper investigation, especially over a long-term period. 

Only a small number of studies have been performed considering the effects of BSF-based diets on the whole life cycle of fish [15,19] and, to our knowledge, none on the progeny. It has been demonstrated that parental diet, especially in terms of *n*-3 and *n*-6 PUFA profile, can affect oocyte composition, embryonic development as well as progeny health in different vertebrate species [20,21,22], including teleost fishes [19,23,24]. In this regard, there is evidence that the environmental factors experienced by parents, including nutrient availability during growth and reproduction, can have long-lasting effects on offspring metabolism [25,26]. In particular, in fish, maternally derived nutrients have direct impacts on the progeny during embryogenesis, endogenous feeding period and beyond yolk exhaustion [27,28,29]. Differently from mammals which encounter fluctuations in maternal nutrients, hormones and metabolites during gestation and lactation the early nutritional environment of fish larvae (from fertilization to yolk absorption) is fixed before fertilization [30]. For that reason, parental feeding is very effective for nutritional programming in fish, showing substantial effects on progeny metabolism, growth, survival and transcriptional profile [31,32,33]. The exposure to dietary stimulus during critical fish life cycle stages can lead to long-term changes in metabolic processes, in a phenomenon called nutritional programming [27]. 

In aquaculture, nutritional programming can be useful to produce fish that are more adapted to the farming conditions and better accustomed to use alternative dietary ingredients lacking specific macronutrients [34,35]. In this regard, it has been demonstrated that partial replacement (60%) of FO with linseed oil in gilthead seabream (*Sparus aurata*) broodstock diets induced long-term persistent effects on the progeny, which showed higher ability to use low FM- and FO-based diets even after 16 months post hatching [36,37]. This adaptation was dependent on the regulation of hepatic lipid metabolism which, in turn, induced positive and persistent changes in the progeny PUFA profile [26,37,38]. In this sense, given the importance of long-chain PUFA for human health [39], as well as the increasingly limited availability of aquafeed ingredients rich in these fatty acids [40], nutritional programming may adjust fish metabolism to maximize the ability of farmed fish to use specific ingredients, with an emphasis on dietary fatty acids [36,37,38].

The present study aimed to investigate, for the first time, whether nutritional programming exists in larval zebrafish fed on diets including increasing dietary levels of full-fat BSF prepupae meal, with emphasis on fish growth, health and fatty acid composition. 

Even though zebrafish is a widely used model organism, information about its dietary predilections and nutritional requirements is mostly unknown [41]. The natural diet of wild zebrafish is composed of a wide variety of benthic and planktonic crustaceans, worms and insect larvae [42]. However, the analysis of wild zebrafish gut contents evidenced that insects, mostly of terrestrial origin, represent their main prey [43,44]. In this regard, a previous study by Vargas and collaborators [45] pointed out that a 100% BSF meal diet did not affect zebrafish larval development over the course of a 21-day experiment. However, in laboratory conditions, zebrafish are known to be regularly fed on commercial diets (i.e., Zebrafeed, Sparos ltd, Olhão, Portugal).

Since zebrafish represents an extraordinary experimental model for aquaculture studies, contributing to our understanding of how mechanisms involved in fish nutrition, welfare and growth take place in farmed fish species [46], testing new dietary formulations with the inclusion of insect meal is necessary for the further development of the aquaculture industry. Finally, due to its complete genome availability and relatively short life cycle, zebrafish allow us to investigate possible dietary effects, and their eventual persistence, throughout the whole life of the fish and across generations, in a relatively short time [31,46,47].

## 2. Materials and Methods

### 2.1. Insect Rearing

The BSF larvae feeding substrate, consisting mainly of coffee silverskin (a coffee industry by-product), was prepared according to Zarantoniello et al. [8]. Briefly, before the feeding substrate preparation, coffee by-product (Saccaria Caffé S.R.L., Marina di Montemarciano, AN, Italy; moisture 44%) was ground in an Ariete 1769 food processor (De’Longhi Appliances Srl, Treviso, Italy) to a particle size of 0.4 ± 0.2 mm. The feeding substrate was formulated including a 10% (w/w) of *Schizochytrium* sp. (provided freeze-dried by AlghItaly Società Agricola S.R.L., Sommacampagna, VR, Italy) to the coffee by-product. To reach a final moisture of ~70% in the feeding substrate [48], distilled water was added. 

Six-day-old BSF larvae (Smart Bugs s.s. Ponzano Veneto, Treviso, Italy) were divided into groups of 640 specimens per replicate (*n* = 65 for a total of 41˙600 BSF larvae). Each replicate consisted of a plastic box (57 × 38 × 16 cm) screened with fine-mesh cotton gauze and covered with a lid with 90 ventilation holes of 0.05 cm diameter [49]. BSF larvae were reared in a climatic chamber at a temperature of 27 ± 1 °C and relative humidity of 65 ± 5% [49], at a density of 0.3/cm^2^ [50], in continuous darkness. The feeding rate per larva was 100 mg/day [51], achievable by adding new feeding substrate once a week (448 g for each box). At the prepupal stage, identified by the change in tegument color from white to black [52], insects were collected, washed, freeze-dried and stored at −80 °C.

### 2.2. Fish Diets

Freeze-dried full-fat BSF prepupae were ground with a Retsch Centrifugal Grinding Mill ZM 1000 (Retsch GmbH, Haan, Germany) for experimental diet preparation. Five experimental diets were prepared as previously described in Zarantoniello et al. [8]. Briefly, diets were formulated to be grossly iso-nitrogenous (50% of CP, N × 6.25, on dry matter) and iso-lipidic (13% on dry matter). A control diet (Hi0) containing fish meal (FM), a vegetable protein mixture (wheat gluten and pea protein concentrates) and fish oil (FO) as major ingredients was prepared and used as the basal diet formulation for the tested BSF-based diets. BSF-based diets were prepared by including graded levels of full-fat BSF prepupae meal (approximatively 25, 50, 75 and 100% named Hi25 and Hi50, Hi75 and Hi100, respectively) in the Hi0 formulation to replace the marine sources (both FM and FO). To maintain the diets’ iso-nitrogenous and iso-lipidic condition, the vegetable protein mixture was adjusted accordingly.

In summary, all the ground ingredients (0.5 mm) were thoroughly blended (Kenwood kMix KMX53 stand Mixer; Kenwood, De’Longhi S.p.a., Treviso, Italy) for 20 min, adding water to obtain an appropriate consistency for pelleting. Pellets were obtained through the use of a 1-mm-die meat grinder, dried at 40 °C for 48–72 h and then ground, sieved and stored in vacuum bags at −20 °C until use. Ingredients and proximate composition of the test diets are reported in Table 1. 

### 2.3. Broodstock Rearing and F0 Production

Zebrafish AB strain adults (broodstock; 1.2 ± 0.4 g), fed on a commercial diet (Blue Line, Macerata, Italy), were laboratory spawned and zebrafish AB embryos (F0) were maintained 48 h in a Techniplast system (Varese, Italy) at 28 °C, pH 7.0, NO_2_ and NH_3_ concentrations < 0.01 mg/L, NO_3_ concentration < 10 mg/L and photoperiod 12 h light/12 h dark. After this first period, embryos were gently collected, counted under a stereomicroscope and randomly assigned to the different experimental groups: F0Hi0, F0Hi25, F0Hi50, F0Hi75 and F0Hi100 (Figure 1). Fish were initially kept in 15 tanks (20 L, three tanks per dietary group with 500 fish per tank), the sides of which were covered with black panels to reduce light reflection [53]. The water in the F0 larval tanks had the same chemical-physical characteristics of the broodstock’s tank and was gently replaced 10 times a day by a dripping system. Starting from 5 days post fertilization (dpf), fish from each dietary group were fed the same experimental diet (Hi0, Hi25, Hi50, Hi75 and Hi00 diets, respectively; 2% body weight) twice a day and, in addition, from 5 to 10 dpf, rotifer *Brachionus plicatilis* (five individuals per mL) were provided to all dietary groups (one feeding in the morning). At 21 days dpf, the required F0 larvae (for details, please see Zarantoniello et al. [8]) from each experimental group (F0Hi0, F0Hi25, F0Hi50, F0Hi75 and F0Hi00, respectively) were sampled, euthanized with a lethal dose of MS222 (1g/L; Merck KGaA, Darmstadt, Germany) and properly stored until further analysis as reported in Zarantoniello et al. [8]. Finally, at 30 dpf, the remaining F0 zebrafish (~200 fish per tank, ~600 per dietary group) were transferred, according to each experimental group, into 15 bigger tanks (80 L; three tanks per dietary group) equipped with mechanical and biological filtration (Panaque, Rome, Italy) until 180 dpf. Feed particle sizes were <100 μm from 5 to 15 dpf, 101–200 μm from 16 to 30 dpf, 201–400 μm from 31 to 60 dpf and 401–600 μm from 61 to 180 dpf. After 180 dpf, adult F0 zebrafish were spawned and progeny embryos (F1; Figure 1) were obtained from each dietary group.

### 2.4. F1 Zebrafish Larvae

First filial generation (F1) embryos, obtained from each experimental group (F1Hi0, F1Hi25, F1Hi50, F1Hi75 and F1Hi00, respectively) were gently collected, counted under a stereomicroscope and transferred to 20 L tanks (three tanks per dietary group). Each experimental group, in triplicate, was composed of 1500 larvae (500 larvae per tank). Starting from 5 to 20 dpf, F1 larvae were fed on the same parental diet twice a day (2% body weight; 100–200 μm particle size) and were named as: (i) F1Hi0 (control) group: F1 larvae fed parental diet with 0% of full-fat BSF prepupae meal; (ii) F1Hi25, F1Hi50, F1Hi75 and F1Hi100 groups: F1 larvae fed parental diet including 25%, 50%, 75% and 100% of full-fat BSF prepupae meal respect to FM, respectively. Furthermore, from 5 to 10 dpf, all groups were fed (one feeding in the morning) on the rotifer *Brachionus plicatilis* (5 ind/mL). All the tanks were siphoned 30 min after feeding (two times a day) to remove possible feed excess and dead larvae which were counted to estimate the survival rate. The required F1 larvae (for details, please see further sections) were sampled at 20 dpf, euthanized with a lethal dose of MS222 (1 g/L; Merck KGaA, Darmstadt, Germany) and properly stored for further analyses.

### 2.5. Biometry

Ten F1 zebrafish larvae (30 per dietary group) were randomly collected from each tank at hatching (3 dpf) and at 21 dpf. Excess water was removed by means of a syringe, and wet weight was determined by an analytical balance (precision: 0.1 mg) by measuring five larva pools at 3 dpf and single specimens at 21 dpf. For each experimental group, specific growth rate (SGR) was calculated as follows: SGR% = (ln*Wf* − ln*Wi*)/t) × 100, where *Wf* is the final wet weight, *Wi*, the initial wet weight, and t, the number of days (17). Survival rate was evaluated by subtracting the number of dead larvae at 21 dpf to the initial number (500 per tank).

### 2.6. Fatty Acid Composition

Experimental diets and F1 zebrafish larvae samples were analyzed (in triplicate) for fatty acid composition according to Zarantoniello et al. [8]. Briefly, samples were minced and homogenized (homogenizer MZ 4110, DCG Eltronic, Monza, Italy), and larvae were also freeze-dried (Edwards EF4, Crawley, Sussex, UK). Aliquots of 200 mg of each sample (three aliquots per sample) were added with 100 μL of Internal Standard (methyl ester of nonadecanoic acid, 99.6%, Dr. Ehrenstorfer GmbH, Augsburg, Germany), and lipid extraction was carried out on lyophilized powders following a microwave-assisted extraction (MAE) [54]. All lipid extracts were evaporated under laminar flow inert gas (N_2_) until constant weight and re-suspended in 0.5 mL of n-epthane. Fatty acid methyl esters (FAMEs) were prepared according to Canonico et al. [55] and were determined using an Agilent-6890 GC System (Milano, Italy) coupled to an Agilent-5973 N quadrupole Mass Selective Detector (MSD) (Milano, Italy) and separated through a CPS ANALITICA CC-wax-MS (30 m × 0.25 mm ID, 0.25 μm film thickness) capillary column [56]. For each analyzed aliquot of sample, at least three runs were performed on the GCMS. 

### 2.7. Histology

Five F1 zebrafish larvae (15 per dietary group) were randomly collected from each tank at 21 dpf, fixed by immersion in Bouin’s solution (Merck KGaA, Darmstadt, Germany) and then stored at 4 °C for 24 h. Samples were washed three times with ethanol (70%) for 15 min and preserved in a new 70% ethanol solution. After dehydration through graded ethanol solution (80%, 95% and 100%), samples were washed with xylene (Bio-Optica, Milano, Italy) and embedded in paraffin (Bio-Optica). Solidified paraffin blocks were cut with a microtome (Leica RM2125 RTS). Sections (5 µm) were stained with Mayer hematoxylin and eosin Y (Merck KGaA; H&E, Darmstadt, Germany) in order to study hepatic parenchyma and intestinal morphology and to measure the perivisceral tissue area or with Alcian blue (Bio-optica) for Alcian blue positive (Ab +) goblet cells detection. Sections were observed using a Zeiss Axio Imager.A2 (Zeiss, Oberkochen, Germany) microscope and images were acquired by a digital camera Axiocam 503 (Zeiss).

To ascertain the degree of hepatic fat accumulation, a quantitative analysis was performed on three section per fish (15 zebrafish larvae per dietary group F1) collected at 50 µm intervals and stained with H&E. The percentage of fat fraction (PFF) was calculated by mean of ImageJ software setting a homogeneous threshold value. Not evaluable areas, such as blood vessels and bile ducts, were not considered. Perivisceral adipose tissue area was measured using ZEN 2.3 software (Zeiss) on three section per fish (15 fish per dietary group) collected at 50 µm intervals and stained with H&E. The semi-quantitative evaluation of histological indexes in the intestine was performed on three transversal sections per fish (15 fish per dietary group; 50 μm intervals) stained with H&E for mucosal folds length and enterocyte supranuclear vacuolization abundance or with Alcian blue for Ab+ goblet cells detection. Specifically, for the morphometric evaluation of mucosal folds height, all the undamaged and non-oblique folds were measured using ZEN 2.3 software (Zeiss). Regarding the semi-quantitative analysis of supranuclear vacuoles and Ab + goblet cells, an arbitrary unit was assigned as described in Panettieri et al. [57]. Scores were assigned as follows: supranuclear vacuoles + = scattered, ++ = abundant; Ab+ goblet cells: + = 0 to 3 per villus, ++ = 4 to 6 per villus, + + + = more than 6 per villus.

### 2.8. Total RNA Extraction and cDNA Synthesis

Total RNA extraction from five F1 zebrafish larvae collected from each tank at 21 dpf (15 per dietary group) was performed using RNAzol RT reagent (Merck KGaA, Darmstadt, Germany) following the manufacturer’s protocol. The final RNA concentration was determined by a NanoPhotometer P-Class (Implen, München, Germany) and the RNA integrity was verified by GelRed^TM^ staining of 28S and 18S ribosomal RNA bands on 1% agarose gel. The cDNA synthesis was performed using the LunaScript RT SuperMix Kit (New England Biolabs, Ipswich, Massachusetts, USA) using 1 µg of total RNA. 

### 2.9. Real-Time PCR

PCRs were performed in an iQ5 iCycler thermal cycler (Bio-Rad, Hercules, CA, USA). Reactions were set on a 96-well plate by mixing, for each sample, 1 µL cDNA diluted 1:10, 5 µL of 2x concentrated iQ^TM^ Sybr Green (Bio-Rad) as fluorescent intercalating agent, 0.3 µM of forward primer and 0.3 µM of reverse primer. The thermal profile for all reactions was 3 min at 95 °C and then 45 cycles of 20 s at 95 °C, 20 s at 60 °C, and 20 s at 72 °C. At the end of each cycle, florescence was monitored, and the melting curve analyses showed in all cases one single peak. Relative quantification of the expression of genes involved in fish growth (insulin-like growth factor 1, *igf1*; insulin-like growth factor 2a, *igf2a;* myostatin, *mstnb*), stress response (glucocorticoid receptor, *nr3c1*; heat shock protein 70, *hsp70.1*), long-chain polyunsaturated fatty acids biosynthesis (fatty acid elongase 2, *elovl2*; fatty acid elongase 5, *elovl5;* fatty acid desaturase 2, *fads2*), appetite response (ghrelin, *ghrl*; neuropeptide y *npy*; cannabinoid receptor 1, *cnr1*; leptin a, *lepa*), immune response (interleukin 1β, *il1b*; interleukin *il10*; tumor necrosis factor a, *tnfa*) and enzymatic hydrolysis of chitin (chitinase 2, *chia.2*; chitinase 3, *chia.3*) was performed. Actin-related protein 2/3 complex, subunit 1A (*arpc1a*) and ribosomal protein, large, 13 (*rpl13*) were used as internal standards in each sample in order to standardize the results by eliminating variation in mRNA and cDNA quantity and quality. Amplification products were sequenced, and homology was verified. No amplification products were observed in negative controls and no primer–dimer formations were observed in the control templates. Data obtained were analyzed using the iQ5 optical system software version 2.0 (Bio-Rad) including GeneEx Macro iQ5 Conversion and GeneEx Macro iQ5 files. The same primer sequences designed using Primer3 (starting from zebrafish sequences available in ZFIN) and reported in Zarantoniello et al. [8] were used in the present study (Table 2). 

### 2.10. Statistical Analyses

All data were analyzed by one-way ANOVA, with diet as the explanatory variable. All ANOVA tests were followed by Tukey’s post-hoc test. The statistical software package Prism5 (GraphPad Software, 6.01 version) was used. Significance was set at *p* < 0.05 and all the results are presented as mean ± SD.

## 3. Results

### 3.1. F1 Zebrafish Larvae—Growth and Survival

Considering SGR%, no significant differences were detected among experimental groups (22.9 ± 1.0, 22.7 ± 0.9, 23.0 ± 1.1, 22.5 ± 0.8 and 22.7 ± 0.9% for F1Hi0, F1Hi25, F1Hi50, F1Hi75 and F1Hi100, respectively). Survival rate did not show significant differences among experimental groups (89 ± 5, 86 ± 6, 85 ± 4, 81 ± 5 and 78 ± 6% for F1Hi0, F1Hi25, F1Hi50, F1Hi75 and F1Hi100, respectively).

### 3.2. F1 Zebrafish Larvae—Fatty Acid Content and Composition

The increasing dietary inclusion level of full-fat BSF prepupae meal resulted in a significant (*p* < 0.05) increase of SFA (Figure 2a). Fish fed BSF-based diets showed significantly (*p* < 0.05) higher percentages of MUFA and *n*-9 compared to F1Hi0 group (Figure 2a). With regard to PUFA content, F1Hi0 and F1Hi25 groups were characterized by a significantly (*p* < 0.05) higher percentage compared to the other experimental groups, which did not show significant differences among them (Figure 2a). Finally, the increasing inclusion levels of BSF prepupae meal in the experimental diets resulted in a significant (*p* < 0.05) dose-dependent *n*-3 decrease and a parallel slight but significant (*p* < 0.05) dose-dependent *n*-6 increase in F1 zebrafish larvae (Figure 2a). Accordingly, the *n*-6/*n*-3 ratio (Figure 2b) evidenced a significant (*p* < 0.05) increase from F1Hi0 to F1Hi100 groups.

Considering the FA composition of F1 zebrafish larvae (Table 3), the most represented SFA in all the dietary treatments was palmitic acid (16:0), followed by stearic (18:0) and myristic (14:0) acids. The percentage of both palmitic and myristic acids was significantly (*p* < 0.05) lower in F1Hi0 group compared to F1Hi50, FiHi75 and F1Hi100 ones, which did not evidence significant differences among them. In addition, the percentage of lauric acid (12:0) significantly (*p* < 0.05) increased according to the increasing BSF prepupae meal dietary inclusion. With regard to MUFA, the predominant fatty acid in all the dietary treatments was oleic acid (18:1n9), which was significantly (*p* < 0.05) higher in all the groups fed BSF-based diets compared to F1Hi0. The increasing dietary BSF prepupae meal level resulted in a significant (*p* < 0.05) increase in both 7-hexadecenoic (16:1n9) and vaccenic (18:1n7) acids percentages, while no significant differences were detected among experimental groups considering palmitoleic (16:1n7) acid. Finally, docosahexaenoic (22:6n3, DHA) and linoleic (18:2n6) acids represented the most abundant PUFA in all the dietary treatments. In particular, linoleic acid levels did not show significant differences among the experimental groups. Conversely, the increasing dietary inclusion level of BSF prepupae meal resulted in a significant (*p* < 0.05) increase in α-linolenic (18:3n3) and arachidonic (20:4n6) acids percentages and in a significant (*p* < 0.05) decrease in DHA and eicosapentaenoic acid (20:5n3, EPA) percentages. Considering the DHA/EPA ratio, F1Hi0 and F1Hi25 groups were characterized by significantly (*p* < 0.05) lower values compared to the other experimental groups which did not evidence significant differences among them. 

### 3.3. F1 Zebrafish Larvae—Histology

With respect to the liver, histological analyses were performed to evaluate lipid accumulation or steatosis. Results evidenced a modest fat liver parenchima with a widespread presence of hepatocytes with cytoplasm filled with fat, interspersed with normal hepatocytes in all the experimental groups highlighting a similar degree of lipid accumulation (Figure 3a–e). These results were confirmed by PFF quantification, which did not show significant differences among the experimental groups (Figure 3f). 

No significant differences were detected among the experimental groups even with regard to the perivisceral adipose tissue area (Figure 3g). 

Finally, with regard to medium intestine (Figure 4a–j), no morphological alterations or signs of inflammation were evident in any of the experimental groups. In addition, no significant differences were observed among the experimental groups in terms of mucosal folds length and supranuclear vacuoles and Ab+ goblet cell abundance (Figure 4k). 

### 3.4. F1 Zebrafish Larvae—Real-Time PCR 

Growth factors. Considering *igf1*, *igf2a* and *mstnb* gene expression (Figure 5a–c), no significant differences were detected among experimental groups.

Stress response. Regarding genes involved in the stress response (*nr3c1* and *hsp70.1*; Figure 5d,e), groups fed the lowest BSF prepupae meal inclusion levels (F1Hi0 and F1Hi25) showed a significant (*p* < 0.05) upregulation compared to F1Hi50, F1Hi75 and F1Hi100, which did not evidence significant differences among them. 

Lipid metabolism. Considering *elovl2* and *elovl5* gene expression (Figure 5f,g), F1Hi100 group shown the highest gene expression (*p* < 0.05) with respect to the other experimental groups, while no significant differences were observed among them (except for *elovl2* gene expression which was significantly downregulated in F1Hi50 compared to F1Hi75; *p* < 0.05). With regard to *fads.2* gene expression (Figure 5h), all groups fed on BSF-based diets showed a significant (*p* < 0.05) upregulation compared to F1Hi0 group. 

Appetite. Regarding gene expression of *ghrl*, *npy* and *cnr1* (Figure 5i–k), the experimental groups fed BSF-based diets showed a significantly (*p* < 0.05) downregulation with respect to F1Hi0 group. With regard to *lepa* gene expression (Figure 5l), a significantly (*p* < 0.05) BSF dose-dependent decreasing trend was evident among the experimental groups, with F1Hi0 that was characterized by a significant upregulation compared to F1Hi75 and F1 Hi100 groups.

Immune response. Considering *il1b* and *il10* gene expression (Figure 5m,n), no significant differences were observed among experimental groups. Differently, groups fed the highest BSF inclusion levels (F1Hi75 and F1Hi100) showed a significant (*p* < 0.05) downregulation in *tnfa* gene expression (Figure 5o) compared to F1Hi0, F1Hi25 and F1Hi50 groups that did not evidence significant differences among them. 

Chitinase. Regarding *chia.2* and *chia.3* gene expression (Figure 5p,q), F1Hi75 and F1Hi100 groups were characterized by a significant (*p* < 0.05) upregulation compared to the other groups which did not show significant differences among them (except for *chia.2* gene expression in F1Hi50 group).

## 4. Discussion

Nutritional programming covers the metabolic adaptations to a dietary stimulus applied in pre- or post-natal stages that persist later in life, possibly improving health and survival [58,59,60]. In this way, parental diet may have effects on the offspring, especially during the early developmental stages characterized by organogenesis, establishment of metabolic pathways and high metabolic plasticity [27]. In the light of FM and FO substitution with more sustainable aquafeed ingredients, several studies have been published highlighting the possibility to nutritionally programming fish offspring via broodstock nutrition to plant-based diets [36,38,61,62,63]; however, no studies have been performed using insect-based diets. 

The present study aimed to evaluate the possible cross-generation effects of BSF-based diets on F1 larval development using zebrafish as an experimental model. Results are discussed with a comparison to F0 zebrafish larvae that were fed on the same BSF-based diets used in the present study, but that were not nutritionally programmed through parental feeding, as reported in Zarantoniello et al. [8]. 

The experimental diets used in in the present study, as well as in Zarantoniello et al. [8], were formulated to be iso-nitrogenous and iso-lipidic, despite a progressive reduction of marine resources and a parallel increase of BSF prepupae meal. To maintain the dietary protein and lipid intake at a constant level, some vegetable ingredients were included. To relate all the results obtained to the dietary BSF prepupae meal inclusion with respect to FM wheat flour and a mixture of highly digestible wheat gluten and pea protein concentrates were used. In this regard, wheat flour is a common dietary filler due to its low nutritional value [64,65]. Furthermore, wheat- and pea protein-based diet have already been demonstrated to not affect zebrafish growth and gene expression compared to a control diet based on FM [66].

In recent years, it has been demonstrated that a nutritional stimulus during broodstock reproduction, represented by up to 70% of FO dietary replacement by a combination of vegetable oils, was able to promote growth performance in gilthead seabream offspring juveniles [36,37,38,67]. Accordingly, the BSF-based diets used in the present study have been shown to improve zebrafish SGR% and to upregulate *igfs* gene expression in F0Hi50, F0Hi75 and F0Hi100 groups compared to a control group [8]. Conversely, the F1 zebrafish larvae analyzed in the present study did not show these differences. This result could be related to the fact that feeding broodstock with experimental diets characterized by different dietary fatty acid profiles can markedly affect offspring lipid metabolism, with particular emphasis on highly energy demanding PUFA biosynthesis [27,36,67,68]. PUFA delivered from parental diet are considered regulators of embryonic gene expression [33,69,70]. As reported by Zarantoniello et al. [8], in F0 zebrafish larvae, an increasing dietary BSF prepupae meal dietary inclusion resulted in a parallel decrease in their PUFA content. In contrast, in the present study, F1 zebrafish larvae from F1Hi50, F1Hi75 and F1Hi100 groups showed a lower PUFA content respect to F1Hi0 and F1Hi25, but they did not evidence significant differences among them. Several studies on farmed fish species reported that providing diets poor in long-chain PUFA to broodstock enhanced offspring ability to synthesize these fundamental FA, as denoted by the upregulation of genes (like *elovl5* and *fads.2*) involved in this process [36,37,38,61,62,71]. In particular, *fads.2* codifies the Δ6-desaturase enzyme, a strong marker for document nutritional programming effects from broodstock to offspring, since it is considered the rate limiting step in long-chain PUFA biosynthesis [72,73,74,75]. As reported in Zarantoniello et al. [8], only the zebrafish larvae fed the highest BSF prepupae meal dietary inclusion (F0Hi100) showed a significant *elovl5* and *fads.2* upregulation compared to the other experimental groups. In the present study, this trend was evident for both *elovl2* and *elovl5*, but not for *fads.2*, expression of which, differently from the F0 study, was significantly higher in all the groups fed BSF-based diets. This upregulation can be correlated with the reduction of differences in PUFA content among zebrafish larvae from F1Hi50, F1Hi75 and F1Hi100 groups. Furthermore, nutritional programming, besides acting on genes involved in lipid metabolism that promote a better use of low FM and FO diets by the offspring, can reduce the risk to develop hepatic steatosis, often evidenced in fish fed both plant- and BSF-based diets [15,16,37]. In this regard, Zarantoniello et al. [8] reported that F0 zebrafish larvae fed the highest BSF dietary inclusion levels (75% and 100%) were characterized by a severe condition of hepatic steatosis that, in turn, was addressed as the potential cause of the overexpression of stress markers (*nr3c1* and *hsp70.1*). Conversely, in the present study, the histological analyses of F1 zebrafish larvae revealed no signs of hepatic steatosis. The same degree of hepatic lipid accumulation was evident among all the experimental groups, despite a dietary BSF dose-dependent increase in *n*-6/*n*-3 ratio that was previously related to steatosis onset [8,16]. Accordingly, the PFF analyses did not show significant differences among the F1 experimental groups, which all evidenced lower values compared to F0Hi75 and F0Hi100 [8]. Furthermore, the expression of genes involved in the stress response (*nr3c1* and *hsp70.1*) was lower in F1Hi50, F1Hi75 and F1Hi100 groups compared to the other ones. These results are in accord with a previous study in which it was demonstrated that stress-related genes can also be modulated in nutritionally programmed offspring from broodstock fed diets with increased substitution of FO with linseed oil [38]. Furthermore, the downregulation of stress markers in F1Hi75 and F1Hi100 could explain the lower leptin (*lepa*) gene expression detected in these groups. In fish, increased cortisol levels result in a synergic increase in hepatic leptin mRNA levels due to the necessity to mobilize energy reserves in response to a stress condition [76,77]. Conversely, this correlation cannot be applicable to F1Hi50 group, which showed a high *lepa* gene expression despite a downregulation of stress markers. However, F1 zebrafish larvae from F1Hi50 were characterized by the highest (even if not significantly) perivisceral adipose tissue area, which could explain the *lepa* gene expression of this group, since the amount of adipose tissue is positively correlated with the circulating leptin levels [78,79].

Considering the orexigenic signals analyzed in the present study, all the groups fed BSF-based diets showed a significant *ghrl*, *npy* and *cnr1* downregulation compared to F1Hi0. These results are in line with the biometric ones and with a previous study in which nutritionally programmed zebrafish were found to be in a satiated state compared to the control groups [80]. Conversely, the higher orexigenic signals gene expression found in all F0 zebrafish larvae fed on BSF-based diets was related to a compensatory mechanism that increased food intake with a consequent faster growth rate, possibly in relation to dietary deficiency of important nutrients, like DHA [8,30].

Nutritional programming may make it possible to obtain fish better adapted to use specific dietary ingredients, also by acting on the gastrointestinal tract which, in fish, is able to adapt to rapid shifts in environmental conditions, including diet [80,81,82]. In the present study, no specific inflammatory events and no differences in histopathological indexes were detected through the histological analyses on F1 zebrafish larvae intestine from any of the dietary treatments. The absence of negative effects on gut health was also observed in F0 zebrafish larvae, which, however, presented an upregulation of molecular markers involved in the immune response when fed on 50%, 75% and 100% BSF FM substitution, possibly suggesting future development of inflammation [8]. Conversely, in the present study, neither *il1b* nor *il10* gene expression showed significant differences among the experimental groups, and *tnfa* was downregulated in F1Hi75 and F1Hi100. Accordingly, it was demonstrated that both anti- and proinflammatory cytokine gene expression can be positively programmed by early nutrition in zebrafish juveniles to better face a dietary challenge later in life [83]. No differences in proinflammatory cytokine gene expression were evidenced also in adult zebrafish fed BSF-based diets (0%, 25% and 50% with respect to FM) over the whole life cycle [15]. The positive effects of BSF-based diets on gut health can be attributed to the properties of lauric acid (12:0) and chitin, which are addressed as immune-boosting molecules [3,84]. The long-term experience with BSF-based diets, potentially also across generations through nutritional programming, could led to a more extended effect of these BSF dietary components, resulting both in the absence of visible inflammatory events in the intestine and to a positive modulation of the molecular markers involved in the immune response. Accordingly, the chitinases (*chia.2* and *chia.3*) upregulation in F1Hi75 and F1Hi100 groups could have possibly increased chitin digestion enhancing its use as prebiotic, having a positive effect on gut microbial communities and, thus, on overall gut health [85,86,87]. 

## 5. Conclusions

The present study highlighted that nutritional programming through broodstock feeding can have positive effects on the offspring when insects are included in the diets. The results demonstrated that, using nutritional programming, the fish meal substitution level with BSF prepupae meal in the diet can be extended by up to almost 100% during zebrafish larval development without negative effects on fish growth and welfare. Nutritional programming should thus be considered as one of the potential solutions for counteracting the recurring negative side effects of high BSF prepupae meal dietary inclusion levels. The results obtained in the present study, which used the experimental model zebrafish, may represent a starting point for their application to finfish culture.

## Figures and Tables

**Figure 1 animals-11-00751-f001:**
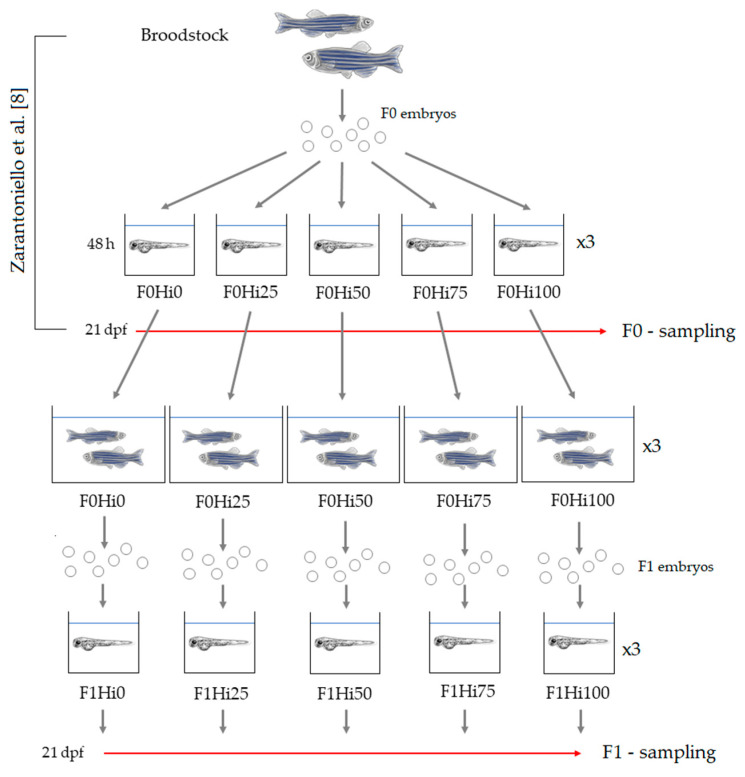
Schematic representation of the experimental design. Fish-fed diets including 0%, 25%, 50%, 75% and 100% of BSF meal respect to FM (F0Hi0, F0Hi25, F0Hi50, F0Hi75 and F0Hi100 for F0 zebrafish and F1Hi0, F1Hi25, F1Hi50, FiHi75 and F1Hi100 for F1 zebrafish larvae). dpf–days post fertilization; F0–parental generation; F1–first filial generation.

**Figure 2 animals-11-00751-f002:**
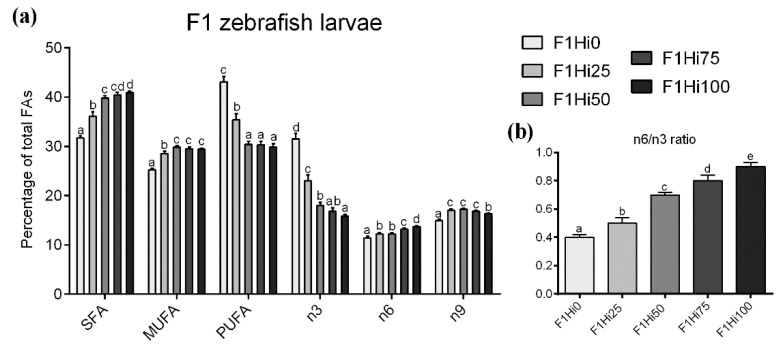
(**a**) Content of saturated fatty acid (SFA), monounsaturated fatty acid (MUFA) and polyunsaturated fatty acid (PUFA) (as % of total fatty acids ) and omega 3 (*n*-3), omega 6 (*n*-6), omega 9 (*n*-9) fatty acid contribution to lipid profile and (**b**) n6/n3 ratio of F1 zebrafish larvae fed the different experimental diets (F1Hi0, F1Hi25, F1Hi50, F1Hi75 and F1Hi100). ^a–e^ Different letters show statistically significant differences among experimental groups compared within the same FA class (*p* < 0.05). Values are reported as mean ± standard deviation (*n* = 3).

**Figure 3 animals-11-00751-f003:**
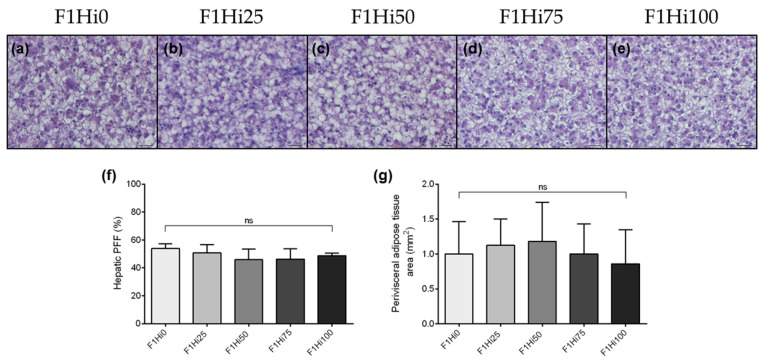
(**a**–**e**) Example of hepatic parenchima histomorphology, (**f**) percentage of fat fraction (PFF) in liver tissue and (**g**) perivisceral adipose tissue area (mm^2^) of F1 zebrafish larvae fed diets including 0%, 25%, 50%, 75% and 100% of BSF prepupae meal respect to FM (F1Hi0, F1Hi25, F1Hi50, F1Hi75 and F1Hi100 groups). Scale bars: 20 μm. For PFF and perivisceral adipose tissue area, values are shown as mean ± standard deviation (*n* = 15). ns: no significant differences.

**Figure 4 animals-11-00751-f004:**
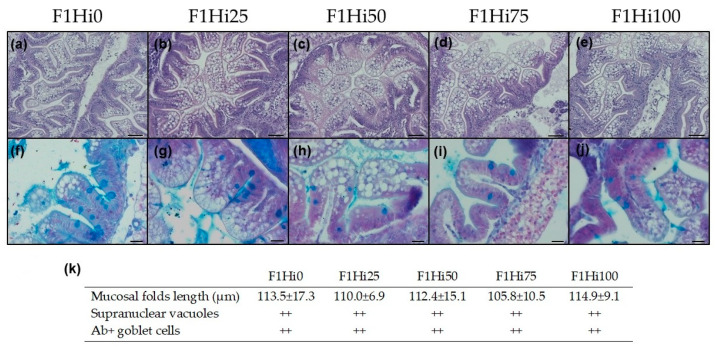
(**a**–**j**) Example of medium intestine histomorphology and (**k**) histological indexes (mucosal folds length, supranuclear vacuoles and Ab+ goblet cell abundance) measured in this gut tract of F1 zebrafish larvae fed diets including 0, 25, 50, 75 and 100% of BSF prepupae meal respect to fish meal (F1Hi0, F1Hi25, F1Hi50, F1Hi75 and F1Hi100 groups). Scale bars and staining: (**a**–**e**) 50 μm, H&E; (**f**–**j**) 20 μm, Ab. Letters: gc = Ab+ goblet cells. For histological indexes (**k**), values of mucosal folds length are shown as mean ± standard deviation (*n* = 15). Scores: supranuclear vacuoles + = scattered, ++ = abundant; Ab+ goblet cells + = 0 to 3 per villus, ++ = 4 to 6 per villus, + + + = more than 6 per villus. No significant differences were detected among the experimental groups.

**Figure 5 animals-11-00751-f005:**
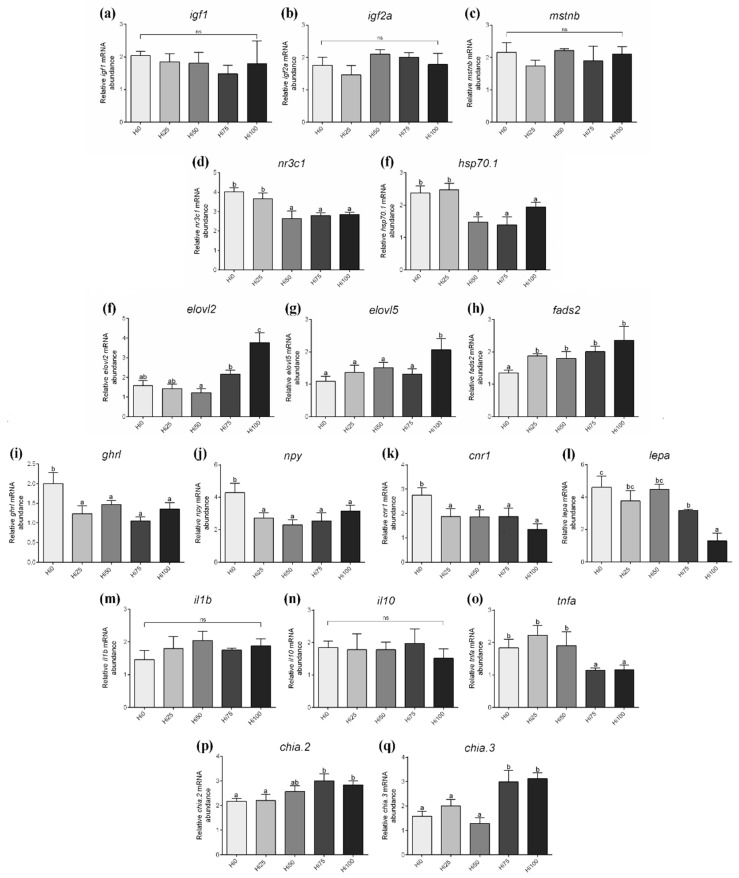
Relative mRNA abundance of genes analyzed in F1 zebrafish larvae fed diets including 0%, 25%, 50%, 75% and 100% of BSF prepupae meal respect to FM (F1Hi0, F1Hi25, F1Hi50, F1Hi75 and F1Hi100 groups). (**a**) *igf1*, (**b**) *igf2a*, (**c**) *mstnb*, (**d**) *nr3c1*, (**e**) *hsp70.1*, (**f**) *elovl2*, (**g**) *elovl5*, (**h**) *fads2*, (**i**) *ghrl*, (**j**) *npy*, (**k**) *cnr1*, (**l**) *lepa*, (**m**) *il1b*, (**n**) *il10*, (**o**) *tnfa*, (**p**) *chia.2* and (**q**) *chia.3*. ^a–c^ Different letters specify statistically significant differences among experimental groups (*p* < 0.05). Values are shown as mean ± standard deviation (*n* = 5). ns: no significant differences.

**Table 1 animals-11-00751-t001:** Ingredients (as g/Kg) and proximate composition (g/100 g) of the experimental diets used in the present study according to Zarantoniello et al. [8].

	Hi0	Hi25	Hi50	Hi75	Hi100
Ingredients (g/kg)					
Fish meal ^1^	470	400	250	110	-
Vegetable protein mix ^2^	220	230	298	385	440
BSF prepupae meal	-	115	235	350	460
Wheat flour ^3^	198	172	120	110	72
Fish oil	80	51	25	10	-
Soy lecithin	8	8	8	11	4
Mineral and Vitamin supplements ^4^	14	14	14	14	14
Binder	10	10	10	10	10
Proximate composition (%)					
Moisture	2.9 ± 0.1	4.2 ± 0.1	5.1 ± 0.1	6.5 ± 0.1	7.3 ± 0.1
Crude protein, CP	51.6 ± 0.1	50.7 ± 2.6	50.4 ± 0.3	51.2 ± 1.5	50.5 ± 3.1
Crude lipid, CL	14.4 ± 0.6	13.1 ± 0.4	12.9 ± 0.4	13.2 ± 0.5	13.0 ± 0.5
Nitrogen-free extract	21.3 ± 0.3	20.8 ± 1.0	20.6 ± 0.5	19.0 ± 0.7	18.5 ± 1.3
Ash	9.8 ± 0.2	11.1± 0.01	11.0 ± 0.00	10.1 ± 0.1	10.7 ± 0.1
Fatty acid content (as % of total FA)					
SFA	27.8 ± 1.3 ^a^	40.9 ± 0.7 ^c^	40.0 ± 2.0 ^c^	35.9 ± 0.7 ^b^	37.6 ± 2.8 ^b^
MUFA	24.7 ± 0.6 ^d^	19.8 ± 0.3 ^b^	19.0 ± 0.9 ^a^	21.5 ± 0.2 ^c^	20.0 ± 1.0 ^b^
PUFA	47.4 ± 1.4 ^c^	39.3 ± 1.0 ^a^	41.0 ± 1.0 ^ab^	42.6 ± 0.3 ^b^	42.4 ± 3.2 ^b^
n3	38.8 ± 1.4 ^e^	27.6 ± 0.9 ^d^	20.8 ± 0.9 ^c^	15.6 ± 0.3 ^b^	11.1 ± 3.1 ^a^
n6	8.6 ± 0.1 ^a^	11.7 ± 0.3 ^b^	20.2 ± 0.4 ^c^	26.9 ± 0.1 ^d^	31.3 ± 0.9 ^e^
n9	13.9 ± 0.3 ^c^	10.7 ± 0.2 ^a^	12.1 ± 0.7 ^b^	14.6 ± 0.2 ^d^	15.2 ± 0.7 ^e^
n6/n3	0.22 ± 0.05 ^a^	0.42 ± 0.10 ^b^	1.00 ± 0.10 ^c^	1.70 ± 0.10 ^d^	2.80 ± 0.20 ^e^

^1^ Raw ingredient kindly supplied by Skretting Italia. ^2^ Vegetable protein mix (pea protein concentrate: wheat gluten, 0.6:1 w/w in all the experimental diets) provided by Lombarda trading srl (Casale Belvedere, CR, Italy) and Sacchetto spa (Lagansco, CN, Italy). ^3^ Consorzio Agrario (PN, Italy); ^4^ Mineral and Vitamin supplement composition (% mix): CaHPO_4_2H_2_O, 78.9; MgO, 2.725; KCl, 0.005; NaCl, 17.65; FeCO_3_, 0.335; ZnSO_4_.H_2_O, 0.197; MnSO_4_.H_2_O, 0.094; CuSO_4_.5H_2_O, 0.027; Na_2_SeO_3_, 0.067; thiamine hydrochloride (vitamin B1), 0.16; riboflavin (vitamin B2), 0.39; pyridoxine hydrochloride (vitamin B6), 0.21; cyanocobalamin (vitamin B12), 0.21; niacin (vitamin PP or B3), 2.12; calcium pantothenate, 0.63; folic acid, 0.10; biotin (vitamin H), 1.05; myo-inositol (vitamin B7), 3.15; stay C Roche (vitamin C), 4.51; tocopherol (vitamin E), 3.15; menadione (vitamin K3), 0.24; retinol (vitamin A 2500 UI/kg diet), 0.026; cholecalciferol (vitamin D3 2400 UI/kg diet), 0.05; choline chloride, 83.99. For proximate composition and fatty acid content, values reported as mean ± SD (*n* = 3). ^a–e^ Different letters show statistically significant differences among experimental groups compared within the same FA class (*p* < 0.05). SFA-saturated fatty acid; MUFA-monounsaturated fatty acid; PUFA-polyunsaturated fatty acid; n3, n6, n9-omega 3, omega 6 and omega 9 fatty acid, respectively.

**Table 2 animals-11-00751-t002:** Primer sequences used in the present study and ZFIN IDs reported in previous studies.

Gene	Forward Primer (5′-3′)	Reverse Primer (5′-3′)	References
*arpc1a*	CTGAACATCTCGCCCTTCTC	TAGCCGATCTGCAGACACAC	[8,16]
*rpl13*	TCTGGAGGACTGTAAGAGGTATGC	AGACGCACAATCTTGAGAGCAG	[8,16]
*igf1*	GGCAAATCTCCACGATCTCTAC	CGGTTTCTCTTGTCTCTCTCAG	[8,16,45]
*igf2a*	GAGTCCCATCCATTCTGTTG	GTGGATTGGGGTTTGATGTG	[8,16,45]
*mstnb*	GGACTGGACTGCGATGAG	GATGGGTGTGGGGATACTTC	[8,16,45]
*nr3c1*	AGACCTTGGTCCCCTTCACT	CGCCTTTAATCATGGGAGAA	[8,16,45]
*hsp70.1*	TGTTCAGTTCTCTGCCGTTG	AAAGCACTGAGGGACGCTAA	[8,16,45]
*elovl2*	CACTGGACGAAGTTGGTGAA	GTTGAGGACACACCACCAGA	[8,16,45]
*elovl5*	TGGATGGGACCGAAATACAT	GTCTCCTCCACTGTGGGTGT	[8,16,45]
*fads2*	CATCACGCTAAACCCAACA	GGGAGGACCAATGAAGAAGA	[8,16]
*ghrl*	CAGCATGTTTCTGCTCCTGTG	TCTTCTGCCCACTCTTGGTG	[8,16]
*npy*	GTCTGCTTGGGGACTCTCAC	CGGGACTCTGTTTCACCAAT	[8]
*cnr1*	AGCAAAAGGAGCAACAGGCA	GTTGGTCTGGTACTTTCACTTGAC	[8,16]
*lepa*	CTCCAGTGACGAAGGCAACTT	GGGAAGGAGCCGGAAATGT	[8,16]
*il1b*	GCTGGGGATGTGGACTTC	GTGGATTGGGGTTTGATGTG	[8,16]
*il10*	ATTTGTGGAGGGCTTTCCTT	AGAGCTGTTGGCAGAATGGT	[8,16]
*tnfa*	TTGTGGTGGGGTTTGATG	TTGGGGCATTTTATTTTGTAAG	[8,16]
*chia.2*	GGTGCTCTGCCACCTTGCCTT	GGCATGGTTGATCATGGCGAAAGC	[8,16,45]
*chia.3*	TCGACCCTTACCTTTGCACACACCT’	ACACCATGATGGAGAACTGTGCCGA	[8,16,45]

**Table 3 animals-11-00751-t003:** Fatty acid composition (as % of total FA) and DHA/EPA ratio of F1 zebrafish larvae.

	F1 Zebrafish Larvae				
	F1Hi0	F1Hi25	F1Hi50	F1Hi75	F1Hi100
10:0	0.010 ± 0.001	0.023 ± 0.001	0.047 ± 0.004	0.052 ± 0.003	0.080 ± 0.009
12:0	0.29 ± 0.01 ^a^	2.70 ± 0.10 ^b^	4.80 ± 0.30 ^c^	5.70 ± 0.30 ^d^	6.40 ± 0.50 ^e^
13:0	0.051 ± 0.002	0.067 ± 0.002	0.084 ± 0.005	0.088 ± 0.002	0.092 ± 0.008
14:0	4.5 ± 0.3 ^a^	5.2 ± 0.3 ^ab^	5.5 ± 0.5 ^b^	5.4 ± 0.5 ^b^	5.5 ± 0.5 ^b^
14:1n5	0.09 ± 0.01	0.17 ± 0.02	0.28 ± 0.03	0.29 ± 0.03	0.26 ± 0.03
15:0	0.78 ± 0.02	0.90 ± 0.03	0.98 ± 0.04	0.98 ± 0.06	0.99 ± 0.07
16:0	18.2 ± 0.8 ^a^	19.6 ± 1.0 ^ab^	20.3 ± 1.0 ^b^	20.2 ± 0.7 ^b^	19.9 ± 0.9 ^b^
16:1n9	0.90 ± 0.05 ^a^	1.10 ± 0.04 ^b^	1.40 ± 0.10 ^c^	1.30 ± 0.10 ^c^	1.30 ± 0.10 ^c^
16:1n7	7.0 ± 0.5 ^a^	7.5 ± 0.4 ^a^	7.5 ± 0.6 ^a^	7.2 ± 0.6 ^a^	7.2 ± 0.5 ^a^
16:2n7	0.19 ± 0.02	0.20 ± 0.01	0.22 ± 0.02	0.29 ± 0.03	0.28 ± 0.03
17:0	0.80 ± 0.04 ^a^	0.90 ± 0.05 ^a^	1.20 ± 0.05 ^b^	1.30 ± 0.07 ^b^	1.20 ± 0.09 ^b^
17:1n7	0.09 ± 0.01	0.29 ± 0.02	0.46 ± 0.04	0.54 ± 0.04	0.54 ± 0.04
18:0	6.6 ± 0.3 ^a^	6.2 ± 0.4 ^a^	6.3 ± 0.5 ^a^	6.1 ± 0.5 ^a^	6.1 ± 0.4 ^a^
18:1n9	12.4 ± 0.6 ^a^	14.4 ± 1.0 ^b^	14.9 ± 1.0 ^b^	14.7 ± 1.2 ^b^	14.6 ± 1.1 ^b^
18:1n7	3.1 ± 0.2 ^a^	3.6 ± 0.2 ^a^	4.4 ± 0.3 ^b^	4.7 ± 0.4 ^bc^	5.1 ± 0.4 ^c^
18:2n6	8.7 ± 0.6 ^a^	9.0 ± 0.5 ^a^	8.1 ± 0.6 ^a^	8.8 ± 0.7 ^a^	9.0 ± 0.6 ^a^
18:3n3	1.3 ± 0.1 ^a^	1.3 ± 0.1 ^a^	1.7 ± 0.1 ^b^	1.9 ± 0.2 ^b^	1.7 ± 0.2 ^b^
20:0	0.32 ± 0.02	0.30 ± 0.02	0.28 ± 0.02	0.29 ± 0.02	0.33 ± 0.03
20:1n9	1.06 ± 0.06	1.01 ± 0.07	0.68 ± 0.04	0.60 ± 0.04	0.40 ± 0.03
20:2n6	0.31 ± 0.03	0.30 ± 0.02	0.30 ± 0.03	0.37 ± 0.04	0.37 ± 0.03
20:3n6	0.42 ± 0.04	0.56 ± 0.05	0.82 ± 0.08	0.92 ± 0.07	1.00 ± 0.10
20:4n6	2.0 ± 0.1 ^a^	2.3 ± 0.1 ^a^	2.9 ± 0.2 ^b^	3.1 ± 0.2 ^bc^	3.3 ± 0.2 ^c^
20:3n3	0.11 ± 0.01	0.10 ± 0.01	0.11 ± 0.01	0.10 ± 0.01	0.11 ± 0.01
20:5n3	8.6 ± 0.5 ^c^	5.0 ± 0.4 ^b^	3.1 ± 0.3 ^a^	2.8 ± 0.3 ^a^	2.7 ± 0.2 ^a^
22:0	0.17 ± 0.02 ^c^	0.24 ± 0.03 ^b^	0.30 ± 0.03 ^a^	0.25 ± 0.03 ^a^	0.28 ± 0.03 ^a^
22:1n9	0.48 ± 0.05	0.47 ± 0.04	0.21 ± 0.03	0.15 ± 0.03	0.03 ± 0.01
22:6n3	21.5 ± 1.0 ^c^	16.6 ± 1.1 ^b^	13.1 ± 1.0 ^a^	12.0 ± 0.9 ^a^	11.3 ± 0.8 ^a^
DHA/EPA	2.5 ± 0.2 ^a^	3.3 ± 0.4 ^b^	4.2 ± 0.5 ^c^	4.3 ± 0.5 ^c^	4.2 ± 0.4 ^c^

Fish fed diets including 0%, 25%, 50%, 75% and 100% of BSF meal respect to fish meal (F1Hi0, F1Hi25, F1Hi50, F1Hi75 and F1Hi100). Means within rows bearing different letters (^a–e^) are significantly different (*p* < 0.05). Values are reported as mean ± standard deviation (*n* = 9). Statistical analysis was performed only for fatty acids > 1%. FA with a percentage < 1% were excluded from any statistical analyses because their concentrations were close to the limit of detection. DHA-docosahexaenoic acid; EPA-eicosapentaenoic acid.

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
