# Peer review of "Possible Dietary Effects of Insect-Based Diets across Zebrafish (Danio rerio) Generations: A Multidisciplinary Study on the Larval Phase"

_animals, 2021, doi:10.3390/ani11030751_

Round 1

Reviewer 1 Report

Current ms is set to study nutritional programming on zebrafish. Zebrafish is suitable species due to its maturation at young age and well know genome. Five levels of insect meal were fed to broodstock, and later on also to their progeny. The hypothesis apparently has been that changes in response parameters observed in the broodstock, should be different at F1 generation due to nutritional programming. Several response parameters were measured, including fatty acid profile, gut histology and stress related gene activation. Broodstock and F1 were fed in triplicates thus allowing statistical analyses.

To my great surprise, no data on F0 is presented, or at least I cannot locate it. Authors go on lengths do give detailed data on F1, and write as follows: “Results are discussed with a comparison to F0 zebrafish larvae which were fed on the same BSF-based diets used in the present study but that were not nutritionally programmed through parental feeding, as reported in Zarantoniello et al. [8].” For the hypothesis to be tested, to my best knowledge F0 data should be presented here, or zebrafish with similar genetic background should have also been fed the five diets, to make conclusions whether broodstock diet influences F1 responses. Otherwise this is a feeding study with graded levels of insect meal. I will not go further in my referee evaluation before this is explained better in the manuscript.

Author Response

Reviewer 1

Current ms is set to study nutritional programming on zebrafish. Zebrafish is suitable species due to its maturation at young age and well know genome. Five levels of insect meal were fed to broodstock, and later on also to their progeny. The hypothesis apparently has been that changes in response parameters observed in the broodstock, should be different at F1 generation due to nutritional programming. Several response parameters were measured, including fatty acid profile, gut histology and stress related gene activation. Broodstock and F1 were fed in triplicates thus allowing statistical analyses.

To my great surprise, no data on F0 is presented, or at least I cannot locate it. Authors go on lengths do give detailed data on F1, and write as follows: “Results are discussed with a comparison to F0 zebrafish larvae which were fed on the same BSF-based diets used in the present study but that were not nutritionally programmed through parental feeding, as reported in Zarantoniello et al. [8].” For the hypothesis to be tested, to my best knowledge F0 data should be presented here, or zebrafish with similar genetic background should have also been fed the five diets, to make conclusions whether broodstock diet influences F1 responses. Otherwise this is a feeding study with graded levels of insect meal. I will not go further in my referee evaluation before this is explained better in the manuscript.

The present study is part of a larger founded research project that lasted 3 years. After testing diets including different BSF meal during the whole life cycle of Zf and on Zf reproduction, the interesting finding of the last years suggested the authors to perform a cross-generation study. Since data on the larval rearing of the F0 generation were already published at the time of this idea, the data of the present MS should be compared to the previous paper. In order to avoid plagiarism, the authors are not able to introduce the past data in the present MS and, for this reason, a comparison is suggested throughout the present MS.

In this sense, the reviewer should evaluate the present MS by comparing previous results which are well discussed in the discussion section.

Reviewer 2 Report

In few places English grammar/ syntax could be corrected. First paragraph of introduction should be separated into sections. Does insect based died has more advantages? Authors should emphasize on this issue in the introduction. Line 149: What was the weight of fish? What were the responses on immune parameters? Whether selected primers were specific to zebrafish? Did authors verify if target genes were amplified properly? Growth performances could be shown in table. Did all the fish were survived the trail duration? Discussion part: Need very careful revision to improve the content. Why these results are important that should be discussed properly in the light of available literature. Table 3: Results are expressed as mean withStandard deviation or standard error? section 3.2: reduce the content slightly.

Author Response

Reviewer 2

In few places English grammar/ syntax could be corrected.

English has been edited.

First paragraph of introduction should be separated into sections.

As requested, the introduction has been split in more paragraphs.

Does insect based died has more advantages? Authors should emphasize on this issue in the introduction.

This aspect was added to the introduction section“In light of the circular economy concept and importance of by-product reuse, insects have gained great attention as promising aquafeed ingredient due to their nutritional value, bio-converting efficiency and their low environmental requirements [4,5].” No much emphasis has been given to this aspect since this is not the main topic of the paper.

Line 149: What was the weight of fish? What were the responses on immune parameters?

The average weight of the broodstock has been added.

Whether selected primers were specific to zebrafish? Did authors verify if target genes were amplified properly?

Yes, they are. The authors are sorry since they forgot to add the specific sentence that they usually add to this section. Primers were designed starting from sequences available in ZFIN (Zebrafish Information Network), which is the zebrafish database of genetic and genomic data. To verify the target genes proper amplification, PCR products were sequenced, and homology was verified (the missing sentence has been added in the Real-time PCR section). These same primers were already used and verified in past studies and, for this reason, aside adding the specific sentence, the reference of a past paper was added to clarify this point.

Growth performances could be shown in table. Did all the fish were survived the trail duration?

As suggested by Reviewer #3, the Figure about SGR% (Fig. 2) has been removed since data are reported in the text. Survival rate was approximatively 80% for all the experimental groups. Mean survival of each experimental group is reported in 3.1. paragraph. Please see sentence: “Survival rate did not show significant differences among experimental groups (89±5, 86±6, 85±4, 81±5 and 78±6 % for F1Hi0, F1Hi25, F1Hi50, F1Hi75 and F1Hi100, respectively).

Discussion part: Need very careful revision to improve the content. Why these results are important that should be discussed properly in the light of available literature.

Even if the discussion section was mainly focused on a comparison between F0 and F1 larval development, the available literature regarding nutritional programming through broodstock nutrition (which is only on the effects of plant-based diets) and on the issues of high dietary BSF prepupae meal inclusion levels (especially in zebrafish) was well considered. The importance of our results, starting to the available literature background, is highlighted in the Conclusion section: “The present study highlighted that nutritional programming through broodstock feeding can have positive effects on the offspring when insects are included in the diets. Results demonstrated that, using nutritional programming, the fish meal substitution level with BSF prepupae meal in the diet can be extended up to almost 100% during zebrafish larval development without negative effects on fish growth and welfare. Nutritional programming should thus be considered as one of the potential solutions to counteract the recurring negative side effects of high BSF prepupae meal dietary inclusion levels. Results obtained in the present study, which used the experimental model zebrafish, may represent a starting point for their application to finfish culture.”

Table 3: Results are expressed as mean with Standard deviation or standard error?

Values are expressed as mean ± SD (the sentence has been added in Table 3 caption).

section 3.2: reduce the content slightly.

The content has been reduced by removing the part about the experimental diets as suggested by Reviewer #3.

Reviewer 3 Report

In addition to the submitted ms Zaratoniello et al. appear to have published several articles about the FM replacement by BSF meal. I regard this as an important topic for the future development of aquaculture industry. Despite the submitted ms being interesting and rather well written, it has certain shortcomings and should thus be improved considerably.  

The ms is a continuation experiment for the published article by the authors in 2020. Because of continuation of the previous experiment, I found somewhat confusing to read this ms along with the published one. Most of all, some parts of the text is apparent self-plagiarism. In addition, the authors have included tables and figures, which already have been published in the 2020 article, and I did not see any permission from Elsevier to republish those data. I suggest you to consider what really needs to be included here and where it is enough to refer to the previous article, and when needed, request the permission from the other publisher.

The authors have not considered at all what zebrafish eats in the wild: they eat insects and zooplankton (e.g. DOI: 10.1111/j.1095-8649.2007.01492.x). As such, selecting zebrafish as a model for a FM replacement study is far from an ideal situation, and in that sense I would see that the study idea should be reversed. The natural diet and it’s potential effects in this study must be taken into account both in the introduction and discussion. In addition, the title of ms looks very strange in this regard, and it will require reconsideration.

The term nutritional programming will need a better description in the introduction.

This experiment was not only about FM replacement by BSF meal but there were also large differences in the diets in regards the amount of protein mix, wheat flour and FO. The authors should explain the reason for these differences, and point out in discussion that what may have been the influence of the other ingredient changes on the results.  

Please justify the use of such a short study period. You were feeding the fish only for 2 weeks and then took the final sample. Is this time sufficiently long for getting reliable results to be used to make conclusions about the effects and usability of these diets?

The paragraph 3.2. starts with “Diets”, but the diets are not a result and this would belong to M&M. However, this is pretty much repetition from the 2020 article and should be removed/shortened.

I attach a marked PDF where I have added numerous minor comments for improvements and changes.

Author Response

Reviewer 3

In addition to the submitted ms Zaratoniello et al. appear to have published several articles about the FM replacement by BSF meal. I regard this as an important topic for the future development of aquaculture industry. Despite the submitted ms being interesting and rather well written, it has certain shortcomings and should thus be improved considerably. The ms is a continuation experiment for the published article by the authors in 2020. Because of continuation of the previous experiment, I found somewhat confusing to read this ms along with the published one.

The authors are sorry for this, but this last experiment was not originally in program. After a 3 years project we observed some interesting results and decided to go further in a F1 study.

Most of all, some parts of the text is apparent self-plagiarism. In addition, the authors have included tables and figures, which already have been published in the 2020 article, and I did not see any permission from Elsevier to republish those data. I suggest you to consider what really needs to be included here and where it is enough to refer to the previous article, and when needed, request the permission from the other publisher.

Authors want to thank for the observation. Tables and Figures already published just referred to the experimental diets and to the primers list which were exactly the same used in the previous study (Zarantoniello et al. [8]). However, to solve the problem we have modified the incriminated figure and tables. In particular, data regarding dietary fatty acid composition were reported in Table 1 (in numeric format) and removed from the figure which reported the same graphs already published in the previous work. In the primers table, we have maintained the sequences, but we have substituted the column “ZFIN ID” with a column “References” to state that sequences were the same used in the previous study. The Ms has been verified for plagiarism through the use of specific programs.

The authors have not considered at all what zebrafish eats in the wild: they eat insects and zooplankton (e.g. DOI: 10.1111/j.1095-8649.2007.01492.x). As such, selecting zebrafish as a model for a FM replacement study is far from an ideal situation, and in that sense I would see that the study idea should be reversed. The natural diet and it’s potential effects in this study must be taken into account both in the introduction and discussion.

The rev. is right. On this regard we published a paper about the use of 100% BSF meal in larval ZF culture (Rearing zebrafish on black soldier fly (Hermetia illucens): biometric, histological, spectroscopic, biochemical and molecular implications.)  A specific sentence was added at the end of the introduction: “Even though zebrafish is a widely used model organism, information about its dietary predilections and nutritional requirements are mostly unknown [41]. The natural diet of wild zebrafish is composed by a wide variety of benthic and planktonic crustaceans, worms and insect larvae [42]. However, the analysis of wild zebrafish gut contents evidenced that insects, mostly of terrestrial origin, represent the main prey [43,44]. On this regard, a previous study by Vargas and collaborators [45] pointed out that a 100% BSF meal diet did not affect zebrafish larval development over a 21 days experiment. However, in laboratory conditions, zebrafish are known to be regularly fed on commercial diets (i.e. Zebrafeed, Sparos ltd, Olhão, Portugal).

In addition, the title of ms looks very strange in this regard, and it will require reconsideration.

The title was partially rewritten and emphasis on the larval phase was added.

The term nutritional programming will need a better description in the introduction.

Nutritional programming has been defined in the introduction section.

This experiment was not only about FM replacement by BSF meal but there were also large differences in the diets in regards the amount of protein mix, wheat flour and FO. The authors should explain the reason for these differences, and point out in discussion that what may have been the influence of the other ingredient changes on the results.  

As pointed out in Materials and Method section: “diets were formulated to be grossly iso-nitrogenous (50% of CP, N x 6.25, on dry matter) and iso-lipidic (13% on dry matter)”.

Wheat flour was used as filler to maintain constant such assumption. This is not unusual (Cardinaletti et al. 2019) due to the low nutritional value of this raw material (crude protein 11.7%; crude lipid 1.2%; crude fiber 1.3%; ash 0.4% on Dry matter; please refers to the Nutrient Requirements of Fish and Shrimp; NRC 2011).

The reviewer also highlights another aspect regarding the amount of other alternative plant protein sources. Both wheat gluten meal (WG) and pea protein concentrate (PPC) are well known to be highly digestible in other fish species (Thiessen et al., 2003; Øverland et al., 2009; Zhang et al., 2012). We selected the pea protein concentrate as this raw ingredient is enhanced in its protein content, reduced the starch content and lowered certain ANF such as tannins compared to whole pea meal by applying dehulling and air classification processes during its manufacturing. Also, wheat gluten meal is considered a well-established alternative protein source for aquafeeds in several farmed fish species (Apper-Bossard et al., 2013), since it has a comparatively high crude protein content if compared to fish meal. The rational of our experimental design was to use such protein concentrate sources to compensate the differences exhibiting between FM (69% CP) and BSM meal (35%).

In addition, only recently, the nutrigenomic effects of single plant proteins (as PPC or WG, included each at 300g/kg) in zebrafish diets on muscle transcriptomic have been published (doi: 10.3389/fgene.2020.575237)

On the other hand, the Reviewer is right when refers to the BSF meal substitutions, since full-fat BSF prepupae meal was used to replace both fish meal and fish oil from the Hi0 formulation.

Accordingly, in the revised version of the MS, it is now reported: “A control diet (Hi0) containing fish meal (FM), a vegetable protein mixture (wheat gluten and pea protein concentrates) and fish oil (FO) as major ingredients was prepared and used as basal diet formulation for the tested BSF-based diets. BSF-based diets were prepared by including graded levels of full-fat BSF prepupae meal (approximatively 25, 50, 75 and 100% named Hi25 and Hi50, Hi75 and Hi100, respectively) in the Hi0 formulation to replace the marine sources (both FM and FO). In order to maintain the diets iso-nitrogenous and iso-lipidic condition, the vegetable protein mixture was adjusted accordingly”.

Please justify the use of such a short study period. You were feeding the fish only for 2 weeks and then took the final sample. Is this time sufficiently long for getting reliable results to be used to make conclusions about the effects and usability of these diets?

According to ZFIN (zfin.org) the zebrafish larval stage starts at protruding mouth stage (72h) and lasts until 21-29 dpf. On this regard, the sampling point was determined to investigate the dietary effects only on the larval development. Furthermore, the experimental diet stimulus was characterized by the exogenous feeding as well as by the endogenous feeding, especially because F0 broodstock were fed on the same experimental diets of the present study during the entire life cycle.  Finally there are already numerous papers on ZF about the effects of nutrition in such a short period of time.

The paragraph 3.2. starts with “Diets”, but the diets are not a result and this would belong to M&M. However, this is pretty much repetition from the 2020 article and should be removed/shortened.

The paragraph 3.2. has been shortened by removing the part about the experimental diets. Data about the dietary fatty acid content have been moved in Table 1.

I attach a marked PDF where I have added numerous minor comments for improvements and changes.

All the minor comments of the PDF have been taken into account. However, the first part of the Figure 1 can’t be removed since it makes the paragraph about experimental design clearer. Without the first part, Figure 1 would represent only a normal feeding study. For this reason the authors wish to maintain this figure.

Round 2

Reviewer 1 Report

I understand the reasoning of not presenting the already published data and accept the comparison to F0-data in the Discussion section. However I remind the difficulties in controlling Type I error in statistical testing, especially when there are a large number of parameters measured. In other words, likelihood of observing some deviations between different studies increases with the number of response parameters measured. Still I admit that the findings of the paper are interesting and should be published.

Authors should, however, note in their manuscript, that while the dietary insect meal changes, so does the levels of plant proteins, and the observation of differences between F0 and F1 is due to change in both insect and plant protein meal levels. This is especially true in the present study, where the diets were produced without heat, often improving the nutrient availability of plant feedstuffs.

In Figure 2 and the related statistical analyses, number of observation should not be the number of individual fish, but the number of replicated tanks (n=3). Individual fish data can be used in nested design. This is a common procedure in aquaculture studies, to avoid pseudoreplication. I understand it is difficult to obey in complicated biocemical analyses, but shuold be used for growth , FCR and similar tank based values.

After these minor changes are made, the ms is publishable.

Author Response

Reviewer 1

I understand the reasoning of not presenting the already published data and accept the comparison to F0-data in the Discussion section. However I remind the difficulties in controlling Type I error in statistical testing, especially when there are a large number of parameters measured. In other words, likelihood of observing some deviations between different studies increases with the number of response parameters measured. Still I admit that the findings of the paper are interesting and should be published.

The authors agree about the possible statistical issues in comparing results from different studies characterized by several parameters analysed. However, the authors think that using a multidisciplinary approach is a proper way to study nutritional responses of fish in order to have a comprehensive overview. Unfortunately, the F1 experiment was not originally in program. After a 3 years project we observed some interesting results and decided to go further in a F1 study. We discussed the results trying to highlight how the same insect-based diets can affect zebrafish larval development in non-programmed or in programmed fish. This comparison allowed to point out the potential advantages of nutritional programming with insect-based diets since, to the best of our knowledge, there are no studies published about this topic. The only ones available for nutritional programming via broodstock nutrition are about plant-based diets and were considered to support our results.  

Authors should, however, note in their manuscript, that while the dietary insect meal changes, so does the levels of plant proteins, and the observation of differences between F0 and F1 is due to change in both insect and plant protein meal levels. This is especially true in the present study, where the diets were produced without heat, often improving the nutrient availability of plant feedstuffs.

This aspect was already pointed out by rev. e and we already provided a pèroper answer that was accepted by the same reviewer.

As pointed out in Materials and Method section: “diets were formulated to be grossly iso-nitrogenous (50% of CP, N x 6.25, on dry matter) and iso-lipidic (13% on dry matter)”.

Wheat flour was used as filler to maintain constant this assumption. This is not unusual (Cardinaletti et al. 2019) due to the low nutritional value of this raw material (crude protein 11.7%; crude lipid 1.2%; crude fiber 1.3%; ash 0.4% on Dry matter; please refers to the Nutrient Requirements of Fish and Shrimp; NRC 2011). Furthermore, both wheat gluten meal (WG) and pea protein concentrate (PPC) are well known to be highly digestible in other fish species. We selected the pea protein concentrate as this raw ingredient has been enhanced in the protein content, reduced the starch content and lowered certain ANF such as tannins compared to whole pea meal by applying dehulling and air classification processes during its manufacturing. Also, wheat gluten meal is considered a well-established alternative protein source for aquafeeds in several farmed fish species (Apper-Bossard et al., 2013), since it has a comparatively high crude protein content respect to fish meal. The rational of our experimental design was to use such protein concentrate sources to compensate the differences exhibiting between FM (69% CP) and BSM meal (35%). In the discussion section, we explained why we have excluded the potential effects of the vegetable portion of the experimental diet by adding the following sentence: “The experimental diets used in in the present study as well as in Zarantoniello et al. [8], were formulated to be iso-nitrogenous and iso-lipidic despite a progressive reduction of marine resources and a parallel increase of BSF prepupae meal. To maintain constant the dietary protein and lipid intake, some vegetable ingredients were included. In order to relate all the results obtained to the dietary BSF prepupae meal inclusion respect to FM wheat flour and a mixture of highly digestible wheat gluten and pea protein concentrates were used. On this regard, wheat flour is a common dietary filler due to its low nutritional value [64,65]. Furthermore, wheat- and pea protein-based diet have already been demonstrated to not affect zebrafish growth and gene expression compared to a control diet based on FM [66].

In Figure 2 and the related statistical analyses, number of observation should not be the number of individual fish, but the number of replicated tanks (n=3). Individual fish data can be used in nested design. This is a common procedure in aquaculture studies, to avoid pseudoreplication. I understand it is difficult to obey in complicated biocemical analyses, but shuold be used for growth , FCR and similar tank based values.

The authors are sorry but there was a mistake in Figure 2 caption, reporting n=9 for F1 zebrafish larvae instead of n=3. The number of replicates has been corrected.

After these minor changes are made, the ms is publishable.

Reviewer 2 Report

Authors satisfactorily revised the manuscript.

Author Response

Thank you.